# FedQAPer: Query Attention Pooling for Dimension Alignment in Federated Non-IID Time-Series Forecasting with Personalized Heads

## Abstract

Federated learning (FL) has shown great promise for time-series forecasting, yet a key challenge in real-world applications is feature heterogeneity. Unlike prior work that assumes uniform feature spaces, we construct a more realistic feature-level non-independent and identically distributed (non-IID) scenario by allocating subsets of features to each client. The number of features varies from 1 up to a defined maximum. We introduce FedQAPer, a novel FL framework that combines Query Attention Pooling (QAP) with FedPer algorithm that uses personalized heads for each client to capture local patterns. QAP projects heterogeneous client feature dimensions into a unified representational space, enabling collaborative backbone training across diverse feature configurations. FedPer transforms these aligned representations back to each client's original feature dimension through personalized heads, achieving both global knowledge integration and local specialization. FedQAPer works for various backbone architectures, including both artificial neural network (ANN) models and spiking neural network (SNN) models. Experiments on multivariate time-series benchmarks demonstrate that FedQAPer effectively handles feature heterogeneity and consistently improves forecasting performance across different backbone models.

## 1 Introduction

Time-series forecasting is fundamental to modern data-driven systems, enabling proactive decision-making in diverse fields such as renewable energy management (Gaboitaolelwe et al. (2023)) and smart grid (Aslam et al. (2021)). For these tasks, deep learning models have demonstrated strong performance. Artificial Neural Networks (ANNs), particularly recurrent and attention-based architectures, have emerged as a dominant paradigm due to their unparalleled ability to capture intricate temporal dependencies and long-range patterns within data (Zhou et al. (2022);Wang et al. (2024c)Nie et al. (2023)). Meanwhile, Spiking Neural Networks (SNNs) offer a distinct, compelling advantage in energy efficiency and suitability for resource-constrained edge devices(Feng et al. (2025);Hu et al. (2025)). By encoding temporal patterns through discrete spike events, SNNs can dramatically reduce energy consumption and computational overhead compared to conventional ANNs(Yu et al. (2024a);Skatchkovsky et al. (2019);Wen et al. (2023)). Recent advances in neuromorphic computing further demonstrate that SNN-based federated learning can maintain competitive forecasting accuracy while achieving significant improvements in energy and communication efficiency(Li et al. (2025);Venkatesha et al. (2021)). Given the complementary strengths of both ANNs and SNNs, both are excellent candidates for various time-series forecasting tasks.

Traditional time series forecasting relies on centralized training paradigms that aggregate all data onto a single server (Xu et al. (2024);donghao & wang xue (2024)). However, this approach raises significant privacy concerns and creates substantial communication overhead, particularly problematic in distributed environments (Abdel-Sater & Hamza (2024)). To address these limitations, federated learning (FL) such as FedAvg (McMahan et al. (2023)) has emerged as a promising solution that enables collaborative model training across decentralized edge devices without sharing raw data, thus preserving privacy and reducing communication costs(Zhao et al. (2018)). Yet, they largely ignore a critical challenge in real-world applications: feature-level non-independent and identically distributed (non-IID), where clients possess different sets of input features. Existing FL frame-

works typically require a uniform feature space, making them ill-suited for such scenarios(Li et al. (2020);Ghosh et al. (2021);Karimireddy et al. (2021)) . This limitation is particularly pronounced in real-world deployments such as heterogeneous IoT devices in smart grids or transportation systems, where devices may have different sensor configurations and data collection capabilities.

To overcome this critical limitation, we introduce FedQAPer, a novel FL framework specifically designed to handle feature-level non-IID data in time-series forecasting. Our method enables clients with diverse feature configurations to collaboratively train a global model while maintaining both privacy and efficiency. Our key contributions are as follows:

- **Realistic feature-level non-IID formulation:** Unlike prior work that assumes identical feature spaces, we explicitly construct a more realistic FL setting where each client has a different number and type of features. This better reflects real-world deployments, such as heterogeneous IoT devices in smart grids or transportation systems.
- **Query Attention Pooling (QAP):** We propose QAP to align heterogeneous client features into a unified latent space. This alignment enables consistent backbone training across clients despite their varying feature dimensions.
- **Personalized Federated Learning via FedPer:** We incorporate the FedPer algorithm(Arivazhagan et al. (2020)), equipping each client with lightweight personalized heads. These heads map the QAP-aligned representations back to the client's original feature space, achieving both global knowledge sharing and local specialization.
- **Backbone-Agnostic Design:** FedQAPer is compatible with a wide range of backbone architectures, including both ANN and SNN models, ensuring flexibility and broad applicability.

## 2 RELATED WORKS

### 2.1 FEATURE DISTRIBUTION IN FEDERATED TIME-SERIES FORECASTING

FL has recently been explored in time-series forecasting, where distributed clients such as sensors, base stations, or smart meters collaboratively learn forecasting models without centralizing raw data (Fekri et al. (2022)). A key challenge in this domain is the inherent non-IID nature of local data. Prior studies typically assume that clients share identical feature spaces, while heterogeneity primarily arises from different in data distribution, sample sizes, or temporal dynamics.

For instance, Fed-TREND (Yuan et al. (2024)) addresses data heterogeneity by treating each feature (or a subset of features) from a single dataset as a distinct client, which then collectively trains various forecasting models. Some approaches may partition a single dataset to match the number of clients, ensuring that each client holds the same set of features(Maher et al. (2025)). Other studies treat an entire dataset as a single client for cross-domain feature heterogeneity. For example, TIME-FFM(Liu et al. (2024a)) uses general time-series datasets like ETT(Zhou et al. (2021) and Electricity(Lai et al. (2018)) as individual clients, while HSTFL(Cai & Liu (2024)) holds a unique type of time series data, with the data types varying among different clients. It is also possible to configure clients with multiple datasets from the same domain, rather than from different domains. FedWindT(Arooj (2024)) considers the dataset from each wind power plant, which contains data on wind speed, temperature, and output, as a separate client to predict the power generation of that plant.These methods implement non-IID conditions by creating differences in the feature characteristics of clients. FedAtt (Thwal et al. (2023)) predicts the next day's closing price returns for each of the 45 publicly traded companies using their stock-related data. Non-IID data can also be achieved by varying the time frequency of data across clients, as seen in works like MVFL(Yang et al. (2025)) and FedForecast(Liu et al. (2023)). However, there is limited research on distributing a different number of features from a single dataset to clients.

### 2.2 SPIKING NEURAL NETWORK IN FEDERATED LEARNING

The integration of SNNs with FL offers a promising direction for streamlined, privacy-preserving AI on resource-constrained edge devices. SNNs' event-driven, sparse communication inherently provides high computational performance and low power consumption. Capitalizing on these advantages, FedLEC (Yu et al. (2025)) presented an effective FL algorithm that utilizes SNNs to mitigate

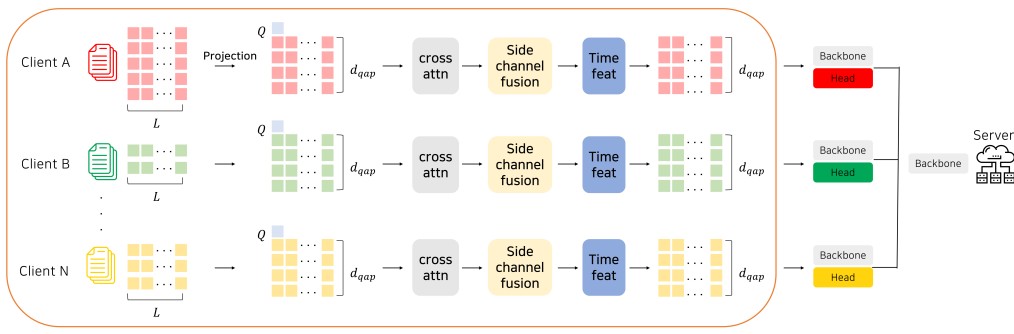

Figure 1: Overview of the proposed FEDQAPER framework.

label skewness and improve convergence in non-IID settings. Xie et al. (2022) demonstrated SNNs' effectiveness in FL for specific appli-cation, such as traffic sign recognition in resource-limited Internet of Vehicles scenarios. Beyond homogeneous model architectures, heterogeneous FL combining convolutional and SNNs has also been explored Yu et al. (2024b). These collective efforts highlight SNNs' significant potential for enabling robust and resource-friendly FL deployments, particularly where computational and communication resources are limited. Despite these advancements, the predominant focus of current SNN-FL research largely remains on classification tasks.

## 3 FEDQAPER

Figure 1 provides an overview of the proposed FedQAPer framework, illustrating the end-to-end pipeline from heterogeneous client inputs through QAP alignment and backbone training to personalized heads for forecasting.

### 3.1 FEATURE ALLOCATION FOR HETEROGENEOUS CLIENT DATA

To simulate feature-level heterogeneity across clients, we construct each client by randomly assigning it a subset of features from a universal dataset, where each feature corresponds to a variate in the multivariate time series. Specifically, given the number of clients $N$ and a maximum feature count $F_{max}$, each client is assigned a unique feature subset. The size of this subset is sampled uniformly, ranging from a single feature up to $F_{max}$ features. This design ensures that the dimensionality of the input space varies across clients. For example, some clients may observe only a single variable, while others access a much richer set of features. Feature assignment is performed independently for each client, without coordination or alignment. This random assignment reflects real-world deployments where local devices or sensors monitor distinct and potentially disjoint subsets of variables, depending on hardware constraints or installation context. Once clients are constructed, each client directly feeds its raw features into the federated model.

### 3.2 QUERY ATTENTION POOLING

In federated time-series forecasting, each client observes a distinct subset of features, leading to heterogeneous input dimensionalities. Let $x^{(i)} \in \mathbb{R}^{B \times L \times F_i}$ denote the input of client $i$ with batch size $B$, sequence length $L$, and feature count $F_i$. Since $F_i$ varies across clients, directly sharing a backbone is infeasible without reconciling dimensional mismatch. We introduce **Query Attention Pooling (QAP)**, a lightweight module that maps arbitrary client features into a unified representational space of dimension $d_{\text{qap}}$. The QAP procedure consists of five steps:

**(1) Value projection with slot embedding.** Each raw feature value at each time step is projected from scalar to $d_{\text{qap}}$ dimensions through a linear transformation, $\text{Linear}(1, d_{\text{qap}})$. Each feature channel is then augmented with a learnable client-specific slot embedding $\mathbf{s}_j^{(i)} \in \mathbb{R}^{d_{\text{qap}}}$ for client $i$ and

feature $j$:

$$\mathbf{v}_{t,j} = \text{Linear}(x_{t,j}^{(i)}) + \mathbf{s}_j^{(i)},$$

where $x_{t,j}^{(i)}$ is the raw value of feature $j$ at time $t$ for client $i$.

**(2) Cross-attention with shared queries.** For each time step independently, the projected feature representations serve as keys and values, while a small set of learnable query vectors $\mathbf{Q} \in \mathbb{R}^{Q \times d_{\text{qap}}}$ (typically $Q = 1$) attend to them through multi-head cross-attention with $h$ heads:

$$\mathbf{z}_t = \text{MultiHeadAttention}(\mathbf{Q}, \{\mathbf{v}_{t,j}\}_{j=1}^{F_i}, \{\mathbf{v}_{t,j}\}_{j=1}^{F_i}).$$

**(3) Statistical side-channel fusion.** To ensure robustness when feature counts vary drastically or when attention becomes unstable, QAP incorporates statistical summaries across the feature dimension at each time step:

$$\mathbf{mean}_t = \frac{1}{F_i} \sum_{j=1}^{F_i} \mathbf{v}_{t,j}, \quad \mathbf{max}_t = \max_{j=1,\ldots,F_i} \mathbf{v}_{t,j}.$$

These are concatenated with the attention output and fused through a two-layer feedforward network:

$$\mathbf{z}_t' = \text{FFN}([\mathbf{z}_t; \mathbf{mean}_t; \mathbf{max}_t]),$$

where $[\cdot; \cdot; \cdot]$ denotes concatenation.

**(4) Query dimension removal.** When using a single query ($Q = 1$), the query dimension is removed, yielding the aligned representation

$$\mathbf{z}_t^{(i)} = \mathbf{z}_t' \in \mathbb{R}^{d_{\text{qap}}}.$$

The complete sequence representation becomes $\mathbf{Z}^{(i)} \in \mathbb{R}^{B \times L \times d_{\text{qap}}}$.

**(5) Time feature integration.** To incorporate temporal context while maintaining dimensional consistency, time features $\mathbf{t}^{(i)} \in \mathbb{R}^{B \times L \times t_{\text{dim}}}$ are linearly projected into the same latent space:

$$\mathbf{t}_{\text{emb}}^{(i)} = \text{Linear}(\mathbf{t}^{(i)}) \in \mathbb{R}^{B \times L \times d_{\text{qap}}}.$$

The QAP-processed features $\mathbf{Z}^{(i)}$ and time embeddings $\mathbf{t}_{\text{emb}}^{(i)}$ are then concatenated and projected to maintain the target dimensionality:

$$\mathbf{Z}_{\text{final}}^{(i)} = \text{Linear}([\mathbf{Z}^{(i)}; \mathbf{t}_{\text{emb}}^{(i)}]) \in \mathbb{R}^{B \times L \times d_{\text{qap}}}.$$

To summarize, we apply Layer Normalization to $\mathbf{v}_{t,j}$ (the value projection with slot embedding) before attention, and dropout to the attention output. Multi-head attention uses $h$ heads with dot-product scaling. The side-channel feed-forward network (FFN) maps the concatenation

$$[\mathbf{z}_t; \mathbf{mean}_t; \mathbf{max}_t] \in \mathbb{R}^{3d_{\text{qap}}}$$

back to $\mathbb{R}^{d_{\text{qap}}}$. After concatenating time embeddings, a linear layer maps from $\mathbb{R}^{2d_{\text{qap}}}$ back to $\mathbb{R}^{d_{\text{qap}}}$, ensuring that the backbone always receives inputs in

$$\mathbb{R}^{B \times L \times d_{\text{qap}}}.$$

This design ensures that both raw feature information and temporal context are jointly encoded in the unified $d_{\text{qap}}$-dimensional space, enabling seamless integration with any backbone architecture. The complete QAP procedure is also summarized in Algorithm 1.

## 3.3 Personalized Federated Learning with FedPer

To integrate the aligned latent representations produced by QAP into federated learning, we adopt **FedPer** (Arivazhagan et al., 2020), a widely used personalization strategy. FedPer separates the model parameters into two groups:

---

**Algorithm 1** Query Attention Pooling (QAP) for Feature-Dimension Alignment

---

**Input:** Client data $x^{(i)} \in \mathbb{R}^{B \times L \times F_i}$, target dimension $d_{\text{qap}}$

**Output:** Aligned representation $z^{(i)} \in \mathbb{R}^{B \times L \times d_{\text{qap}}}$

1. **Value projection + slot embedding:**
   $V = \text{LayerNorm}(W_v(x^{(i)}) + E_i)$
2. **Flatten for attention:**
   $x_{\text{flat}} = \text{reshape}(V, [B \cdot L, F_i, d_{\text{qap}}])$
3. **Cross-attention pooling:**
   $z = \text{Attention}(Q, x_{\text{flat}}, x_{\text{flat}})$
4. **Statistical fusion:**
   $\mu = \text{mean}(x_{\text{flat}}), \; m = \max(x_{\text{flat}})$
   $z = \text{MLP}([z; \mu; m])$
5. **Restore temporal structure:**
   $z^{(i)} = \text{reshape}(z, [B, L, d_{\text{qap}}])$

**return** $z^{(i)}$

---

**Notation:** $B$ = batch size, $L$ = sequence length, $F_i$ = number of features for client $i$, $d_{\text{qap}}$ = latent dimension after QAP alignment, $W_v$ = value projection, $E_i$ = slot embeddings, $Q$ = shared query vector.

---

**(1) Shared backbone**—trained collaboratively across all clients through federated aggregation.

**(2) Personalized heads**—remain local to each client and are never transmitted to the server.

In our setting, once QAP maps a client's raw input with feature dimension $F_i$ into a unified latent space of size $d_{\text{qap}}$, the shared backbone processes this aligned representation. Importantly, because QAP already ensures consistent dimensionality across clients, the output can be directly fed into a wide range of backbone architectures (e.g., transformers, mixers, spiking networks) without requiring any additional preprocessing layers. Each client then applies its own lightweight prediction head to transform the backbone output back to its original feature dimension $F_i$.

During training, backbone parameters and shared QAP components (queries, value projection, attention layers, fusion networks, normalization, time feature projection, and concatenation fusion) are uploaded to the server and aggregated. Client-specific components—including slot embeddings and prediction heads—remain local, ensuring personalized adaptation to each client's unique feature space and output requirements. The federated training procedure with QAP-aligned inputs and FedPer is detailed in Algorithm **??**.

## 4 EXPERIMENTS

We evaluate FedQAPer on two widely used multivariate time-series benchmarks: **Electricity** (321 features) and **Traffic** (862 features). Both datasets are publicly available through the extensive repository of Wang et al. (2024b). Importantly, their high-dimensional feature spaces provide a sufficiently large number of variables to construct diverse clients, making them particularly suitable for our feature-level non-IID experimental design. For our experiments, we set the input sequence length (look-back window) to 96 time steps, and evaluate forecasting performance under three horizons: 48, 96, and 192 steps. The federated learning setup involves 20 clients, and each client receives up to 20 features from the total feature space. All clients participate in every global communication round, ensuring full federation. To ensure reproducibility, we fix the random seed to 42. Each dataset is split into 80% training, 10% validation, and 10% testing. For feature alignment, we apply QAP with query count=1 and latent dimension $d_{\text{qap}}$=128. As backbones, we evaluate both ANN-based models: iTransformer(Liu et al. (2024b)), TimeMixer(Wang et al. (2024a)), DLinear(Zeng et al. (2022)) and SNN-based models: Spikeformer, SpikeRNN (Lv et al. (2024)) . For SNN models, we apply SNN time step=4.

---

**Algorithm 2** Federated Training with FedPer using QAP

**Input:** Global epochs $G$, local epochs $E$, learning rate $\eta$, clients $\mathcal{C}$
**Output:** Shared parameters $\theta^s$, personalized parameters $\{\theta_i^p\}_{i \in \mathcal{C}}$

**Initialization:**
  $\theta^s = \{\theta^{\text{back}}, \theta^{\text{qap-shared}}, \theta^{\text{time}}\}$                                   ▷ Shared parameters
  $\theta_i^p = \{\theta_i^{\text{slot}}, \theta_i^{\text{head}}\}$                                   ▷ Personalized parameters per client

**for** round $t = 1$ to $G$ **do**
    Sample subset $\mathcal{S}_t \subseteq \mathcal{C}$
    Server broadcasts $\theta^s$ to clients $i \in \mathcal{S}_t$
    **for all** client $i \in \mathcal{S}_t$ **do in parallel**
        $\theta_i \leftarrow \theta^s \cup \theta_i^p$                                   ▷ Combine shared and personalized params
        **for** epoch $e = 1$ to $E$ **do**
            **for** batch $(x_i, t_i, y_i) \in \mathcal{D}_i$ **do**
                **1. Forward pass with QAP:**
                $z_i = \text{QAP}(x_i; \theta^{\text{qap-shared}}, \theta_i^{\text{slot}})$
                $z_i^{\text{fus}} = z_i + \text{Linear}(t_i; \theta^{\text{time}})$
                $\hat{y}_i = \text{Head}(\text{Backbone}(z_i^{\text{fus}}; \theta^{\text{back}}); \theta_i^{\text{head}})$
                **2. Update parameters:**
                $\theta_i \leftarrow \theta_i - \eta \nabla \mathcal{L}(\hat{y}_i, y_i)$                                   ▷ Update all params locally
            **end for**
        **end for**
        Send updated $\theta_i^s$ (subset of $\theta_i$) to server                                   ▷ Keep $\theta_i^p$ local
    **end for**
    **Server Aggregation:**
    $\theta^s \leftarrow \sum_{i \in \mathcal{S}_t} w_i \cdot \theta_i^s$                                   ▷ FedAvg on shared parameters only
**end for**

**Notation:** $\mathcal{S}_t$ = sampled clients, $\mathcal{D}_i$ = client $i$'s data, $w_i = |\mathcal{D}_i| / \sum |\mathcal{D}_j|$.
**Shared parameters:** Backbone, Time proj, QAP modules (value_proj, queries, attn, fuse, norm).
**Personalized parameters:** Client-specific Slot Embeddings, Prediction Head.

---

## 4.1 MAIN RESULTS

As shown in Table 1, FedQAPer consistently outperforms baseline approaches across all backbones and datasets. In particular, the method achieves the best or second-best results in almost every setting, showing both strong accuracy and robustness to different horizons and model architectures.

**Metrics.** Since clients possess heterogeneous output feature dimensions, direct aggregation of raw prediction tensors is structurally infeasible. We therefore compute global metrics using dataset-size weighted averaging of locally computed scalar metrics. Specifically, each client $i$ first computes its local MAE averaged over all dimensions, and the server aggregates these as:

$$\text{MAE}_{\text{global}} = \sum_{i=1}^{N} \text{MAE}_i \times \frac{|\mathcal{D}_i|}{\sum_{j=1}^{N} |\mathcal{D}_j|}$$

where $|\mathcal{D}_i|$ denotes the size of client $i$'s local dataset. All metrics are computed on a normalized scale using a StandardScaler fitted on each training split.

On the Traffic dataset, FedQAPer combined with iTransformer delivers the lowest error rates, achieving a top performance with an average MSE of 0.378. For the Electricity dataset, the framework paired with DLinear achieves an impressive result with an average MSE of 0.182. This demonstrates FedQAPer's ability to find the most effective model backbone for diverse datasets and maximize its performance. Notably, our framework is proven to be applicable to both ANN and SNN backbones. While ANN models generally exhibit lower error rates and superior performance, the SNN models, although showing slightly higher errors, demonstrate very competitive performance when considering their potential for optimization on neuromorphic hardware.

Overall, FedQAPer consistently improves prediction accuracy compared to baselines, demonstrating that feature-level alignment through QAP combined with personalized heads provides an effective and versatile solution for federated time-series forecasting.

Table 1: Main results on Electricity and Traffic. Values are **global test** metrics (MSE, MAE). Lower is better.

| Model | | SNN | | | | | | ANN | | | | | |
|---|---|---|---|---|---|---|---|---|---|---|---|---|---|
| | | iSpikeformer | | SpikeRNN | | Spikeformer | | iTransformer | | TimeMixer | | DLinear | |
| Metrics | | MSE | MAE | MSE | MAE | MSE | MAE | MSE | MAE | MSE | MAE | MSE | MAE |
| Traffic | 48 | 0.371 | 0.350 | 0.380 | 0.340 | 0.384 | 0.356 | 0.358 | **0.332** | **0.356** | 0.333 | 0.372 | 0.348 |
| | 96 | 0.401 | 0.359 | 0.401 | 0.354 | 0.406 | 0.358 | 0.387 | **0.343** | 0.390 | 0.348 | **0.340** | 0.359 |
| | 192 | 0.416 | 0.370 | 0.398 | 0.352 | 0.428 | 0.376 | 0.392 | 0.340 | **0.391** | **0.338** | 0.398 | 0.350 |
| | Avg | 0.396 | 0.360 | 0.392 | 0.349 | 0.409 | 0.365 | **0.378** | **0.337** | 0.380 | 0.341 | 0.391 | 0.352 |
| Electricity | 48 | 0.229 | 0.337 | 0.219 | 0.331 | 0.232 | 0.339 | 0.216 | 0.323 | 0.220 | 0.325 | **0.169** | **0.281** |
| | 96 | 0.244 | 0.347 | 0.221 | 0.331 | 0.247 | 0.353 | 0.215 | 0.323 | 0.216 | 0.322 | **0.180** | **0.292** |
| | 192 | 0.231 | 0.340 | 0.241 | 0.341 | 0.251 | 0.359 | 0.229 | 0.333 | 0.219 | 0.327 | **0.198** | **0.309** |
| | Avg | 0.235 | 0.342 | 0.227 | 0.334 | 0.244 | 0.350 | 0.220 | 0.326 | 0.219 | 0.325 | **0.182** | **0.294** |

## 4.2 INFERENCE ENERGY ANALYSIS

To validate the energy efficiency of SNN models, we compare the inference energy consumption between the proposed iSpikeformer and the ANN-based iTransformer model. Inference energy is calculated based on operation energy units at the 45nm technology node. MAC (Multiply-Accumulate) operations consume 4.6 pJ, while AC (Accumulate) operations consume 0.9 pJ.

The inference pipeline of iSpikeformer consists of three stages. In the pre-backbone stage, QAP, Slot Embedding, and Time Projection are performed with float operations, consuming MAC energy. In the backbone stage, spike generation by LIF neurons, sparse matmul operations in SSA (Spiking Self-Attention), and MLP blocks are executed with AC operations proportional to the firing rate. In the final head stage, MAC operations are performed again for mean pooling and final prediction. The key energy savings of SNNs stem from the firing rate. While ANNs perform MAC operations for all computations, SNNs perform AC operations only when spikes are fired, achieving energy efficiency through sparsity.

For a fair apple-to-apple comparison, we exclude the pre-backbone and head portions as they are identical for both models, and compare only the backbone energy.

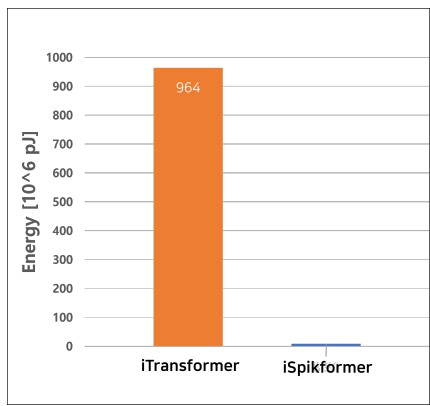

Figure 2: Inference energy comparison between iTransformer and iSpikeformer

The experimental results show that the backbone of iSpikeformer consumes $8.51 \times 10^8$ pJ, which is approximately $113\times$ lower than **iTransformer**'s $9.64 \times 10^{10}$ pJ. This empirically demonstrates that the spike-based sparse computation of SNNs is advantageous for energy-efficient deployment on edge devices.

### 4.3 ABLATION STUDY

#### 4.3.1 EFFECTIVENESS OF FEDERATED LEARNING

To isolate the effect of FedPer, we compare federated training, against a *local-only* setting, where each client trains its own model independently for 100 epoch with an early-stopping patience of 20. For local-only results, we report the average test performance across all clients' best models. All results are obtained under a look-back window of 96 and a prediction horizon of 48 time steps.

As shown in Table 2, FedPer consistently outperforms local-only training across both datasets. On the **Traffic** dataset, the largest gain is observed with TimeMixer, where FedPer reduces errors by approximately 11% in MSE and 17% in MAE.On the **Electricity** dataset, iTransformer under FedPer reduces MSE by about 33% and MAE by nearly 20% compared to local-only training. For SNN backbones, FedPer also delivers consistent improvements, with error reductions in the range of 2–10% compared to local-only training. Overall, these results demonstrate that FedPer not only preserves the strengths of each backbone but also provides clear advantages over isolated local training in feature-level non-IID federated scenarios.

Table 2: Ablation study: federated vs local-only performance (mean $\pm$ std). Lower is better; best of each pair is highlighted in red bold.

| Model | | Traffic | | | | Electricity | | | |
|---|---|---|---|---|---|---|---|---|---|
| | | MSE | | MAE | | MSE | | MAE | |
| | | Fed | Local | Fed | Local | Fed | Local | Fed | Local |
| SNN | iSpikeformer | **0.379** | 0.414 | **0.350** | 0.396 | 0.229 | **0.224** | **0.337** | 0.341 |
| | SpikeRNN | **0.379** | 0.443 | **0.339** | 0.391 | **0.221** | 0.229 | **0.331** | 0.337 |
| | Spikeformer | **0.384** | 0.479 | **0.356** | 0.414 | 0.232 | **0.228** | 0.339 | **0.336** |
| ANN | iTransformer | **0.358** | 0.692 | **0.332** | 0.603 | **0.216** | 0.230 | **0.323** | 0.334 |
| | TimeMixer | **0.356** | 0.476 | **0.333** | 0.433 | 0.220 | **0.218** | **0.325** | 0.330 |
| | DLinear | **0.372** | 0.612 | **0.348** | 0.542 | **0.169** | 0.212 | **0.281** | 0.320 |

#### 4.3.2 COMMUNICATION EFFICIENCY ANALYSIS

We analyze the communication efficiency of FedPer compared to conventional federated learning approaches such as FedAvg. Table 3 presents the parameter distribution of our framework.

In FedPer, shared parameters (QAP and backbone) account for 578,836 (48.4%), while local parameters (forecast head) account for 616,160 (51.6%). Notably, the forecast head constitutes over half of the total model parameters.

In time series forecasting, the forecast head maps from the hidden dimension to the output predictions, and its parameter count scales with $\mathcal{O}(C^2)$, where $C$ denotes the number of features per client. This is because the head must handle client-specific feature dimensions for prediction.

In contrast, FedPer only communicates the shared backbone and QAP, whose parameter counts remain constant regardless of client feature counts, resulting in $\mathcal{O}(1)$ communication cost with respect to $C$.

This architectural choice provides substantial advantages:

- **Reduced communication overhead:** Only 48.4% of parameters are transmitted during federation, while the largest component (forecast head) remains local.

- **Scalability:** As feature heterogeneity increases across clients, the communication cost gap between FedAvg and FedPer widens significantly.

- **Edge device suitability:** The reduced bandwidth requirement enables FedPer to operate effectively on resource-constrained edge devices with limited connectivity.

Table 3: Parameter distribution across components in our framework.

| Component | Parameters | Proportion |
|---|---|---|
| QAP | 167,296 | 14.0% |
| iSpikeformer Backbone | 412,820 | 34.5% |
| Forecast Head | 614,880 | 51.5% |
| **Total** | **1,194,996** | **100%** |

These characteristics make FedPer particularly well-suited for federated time series forecasting scenarios with heterogeneous feature counts, limited bandwidth, and predictable communication budgets.

### 4.3.3 EFFECTIVENESS OF QUERY-ATTENTION POOLING

Also, to **evaluate the effectiveness of QAP, we compare it against a simple linear projection** that only aligns different feature dimensions to a same size. Experiments are conducted on the Traffic dataset. As shown in Table 4, QAP significantly outperforms the linear projection baseline. This demonstrates that QAP not only aligns heterogeneous feature dimensions across clients but also effectively captures important temporal patterns through attention mechanisms. Furthermore, the side channel fusion enables QAP to learn peak and average patterns, leading to improved time series representation learning.

Table 4: Projection vs QAP comparison across different horizons (mean metrics).

| Horizon | MSE | | MAE | |
|---|---|---|---|---|
| | Projection | QAP | Projection | QAP |
| 48 | 1.0053 | **0.4832** | 0.7045 | **0.4172** |
| 96 | 0.9216 | **0.5663** | 0.6704 | **0.4574** |
| 192 | 1.1403 | **0.7109** | 0.7604 | **0.5433** |

### 4.4 CLIENT FEATURE DISTRIBUTION

To better understand the experimental environment, we visualize the client feature allocation under seed=42. Figure 3 and Figure 4 present two perspectives: the number of features assigned to each client (left) and the pairwise overlap of features across clients (right), for the Electricity and Traffic datasets, respectively. The distributions confirm that our construction enforces completely random feature allocation without any coordination. In both datasets, clients are heterogeneous, each receiving different feature subsets with limited overlap. The Traffic dataset exhibits an extreme case of heterogeneity: several clients are assigned nearly the maximum of 20 features, while others have fewer than five. This stark imbalance occurs because the Traffic dataset has a much larger pool of features (862 total), making it more likely for random sampling to create wide disparities across clients. In contrast, the Electricity dataset, which contains fewer total features (321), produces a comparatively milder but still heterogeneous distribution, with most clients falling between 4 and 12 features and only partial overlap. These observations show that the proposed setup consistently produces heterogeneous clients across datasets, with Traffic serving as a particularly challenging benchmark due to its higher intrinsic feature dimensionality. This confirms that our experimental environment accurately simulates the realistic and difficult non-IID conditions encountered in federated learning.

## 5 CONCLUSION

In this work, we presented FedQAPer, a novel federated learning framework that addresses the critical challenge of feature-level heterogeneity in time-series forecasting. Unlike existing FL methods that assume uniform feature spaces across clients, our approach enables effective collabora-

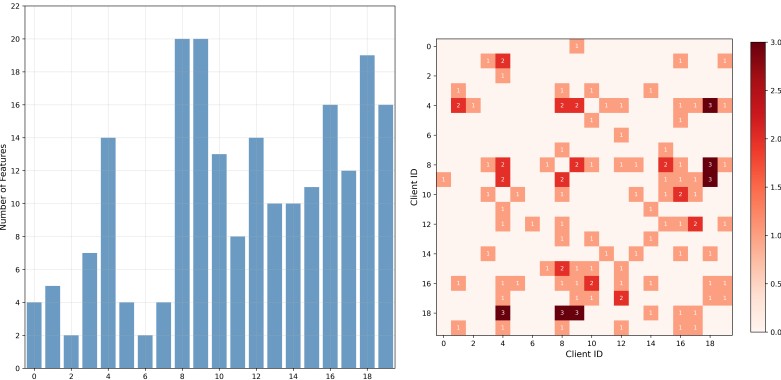

Figure 3: **Electricity** (seed=42). Left: features per client. Right: pairwise feature overlap. Random allocation yields heterogeneous client feature spaces.

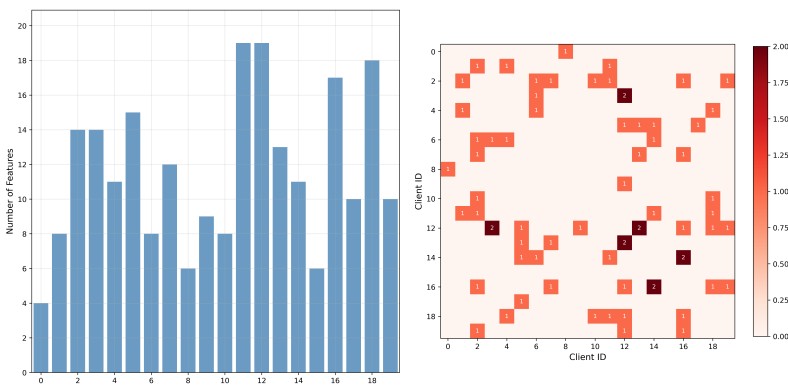

Figure 4: **Traffic** (seed=42). Left: features per client. Right: pairwise feature overlap. Random allocation yields heterogeneous client feature spaces.

tion among clients with diverse feature configurations through the combination of Query Attention Pooling (QAP) and personalized federated learning via FedPer. Our key contributions demonstrate that feature heterogeneity, a common but underexplored challenge in real-world federated deployments, can be effectively managed through dimensional alignment and personalization. The QAP mechanism successfully projects heterogeneous client features into a unified latent space, enabling collaborative training of a shared backbone despite varying input dimensions. Meanwhile, the Fed-Per strategy with personalized heads preserves each client's ability to capture local patterns specific to their unique feature configurations. Experimental results on the Electricity and Traffic datasets confirm that FedQAPer consistently outperforms both baseline federated approaches and local-only training across multiple backbone architectures, achieving improvements of **up to 33%** in MSE and MAE. The framework's compatibility with both ANN and SNN models demonstrates its generality and practical applicability. Overall, FedQAPer represents a significant step toward practical federated learning for time-series forecasting in heterogeneous environments, bridging the gap between theoretical FL frameworks and real-world deployments where feature diversity is the norm rather than the exception. For future work, we aim to further enhance FedQAPer's robustness and flexibility, exploring improved attention variants and more adaptable backbone-head designs to extend its applicability across even more heterogeneous federated scenarios.

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

## A APPENDIX

### A.1 DATASET AND METRIC DETAILS

**Datasets details** The details of the datasets used in the main experiment are shown in Table 5. The dataset size in table is organized in (Train, Validation, Test).

Table 5: Dataset detailed descriptions

| Dataset | Dimension | Frequency | Window length | Horizon length | Dataset size |
|---|---|---|---|---|---|
| Traffic | 862 | Hourly | 96 | {48, 96, 192} | (14035, 1754, 1754) |
| Electricity | 321 | Hourly | 96 | {48, 96, 192} | (21043, 2630, 2631) |

**Metric details** We utilize the mean square error (MSE) as loss function and mean absolute error (MAE) for evaluation. The calculations of these metrics are :

$$\text{MSE} = \frac{1}{N} \sum_{i=1}^{N} \left( X_i - \hat{X}_i \right)^2 \qquad (1)$$

$$\text{MAE} = \frac{1}{N} \sum_{i=1}^{N} \left| X_i - \hat{X}_i \right| \tag{2}$$

where $X, \hat{X} \in \mathbb{R}^{H \times F}$ denote the ground truth and prediction tensors over horizon $H$ time steps with $F$ features. The metrics are calculated over all $N = H \times F$ prediction elements.

## A.2 Hyperparameter Setting

**Backbone Hyperparameter**

Table 6 summarizes the backbone architectures and training-related hyperparameters used in our experiments, including a batch size of 128, a learning rate of $1 \times 10^{-3}$, and the Adam optimizer.

Table 6: Model architecture hyperparameters. Model-specific parameters are listed in the rightmost column.

| Model | Hidden Dim | Layers | Attention Heads | $d_{ff}$ | Kernel Size | Model-specific parameters |
|---|---|---|---|---|---|---|
| iTransformer | 512 | 2 (Encoder) | 8 | 2048 | - | Dropout: 0.1, Activation: gelu |
| TimeMixer | 64 | 2 (PDM blocks) | - | 256 | - | Down-sampling: 2, Moving Avg: 25, Top-K: 5 |
| DLinear | - | - | - | - | - | Moving Avg Kernel: 25, Individual: False |
| SpikeRNN | 64 | 2 (RNN) | - | - | 3 | Steps: 4, Tau: 2.0, LIF Threshold: 1.0 |
| Spikformer | 64 | 2 (Transformer) | 8 | 256 | - | Steps: 4, QK Scale: 0.125, LIF Threshold: 1.0 |
| iSpikeformer | 128 | 2 | 8 | 512 | 3 | tau: 2.0, Surrogate: ATan, Step: multi-step |

**QAP Hyperparameter**

For the query attention pooling (QAP) module, we set the number of queries to $q = 1$. This choice is motivated by two considerations. First, the sequential nature of time-series data inherently contains continuous temporal dependencies, which can be effectively captured with a single query. Increasing the number of queries provides diminishing returns for modeling temporal continuity in forecasting tasks. Second, reducing the number of queries significantly improves computational efficiency—a critical factor in federated learning where both communication and computation costs scale with the number of participating clients. Hence, $q = 1$ strikes an optimal balance between preserving essential temporal information and minimizing resource overhead in federated settings.

To determine the latent dimension of QAP, we conducted a controlled experiment using the iTransformer backbone, which consistently outperformed other candidates under our baseline setting (batch size = 128, lookback window = 96, horizon = 48, global epochs = 100, local epochs = 1, number of clients = 20, and maximum feature count = 20). We varied $d_{\text{qap}}$ across $32, 64, 128, 256$ and observed the validation loss. As shown in Figure 5, the loss remained stable for smaller dimensions but increased significantly at 256. The best performance was achieved at $d_{\text{qap}} = 128$, which we therefore adopt as our default setting.

## A.3 Effect of Maximum Feature Size

Traffic dataset results under different maximum feature sizes with input length 96, horizon 48, and QAP hidden dimension $d_{\text{qap}} = 64$ are shown in Table 7. Bold numbers indicate better performance between $F_{\text{max}} = 20$ and $F_{\text{max}} = 30$. Increasing the number of features per client consistently improves forecasting accuracy across both ANN and SNN backbones, highlighting the benefit of leveraging richer feature contexts in federated learning.

## A.4 Visualization of QAP Transformation

To visualize the effect of QAP, we present an example using the first client from the Electricity dataset, which contains four features. As shown in Figure 6, the raw input time series with four features is transformed by QAP into a 128-dimensional representation ($d_{\text{qap}} = 128$). Since visualizing all 128 dimensions is impractical, we show two representative dimensions to illustrate how QAP captures and transforms the temporal patterns from the original features into a unified representation space.

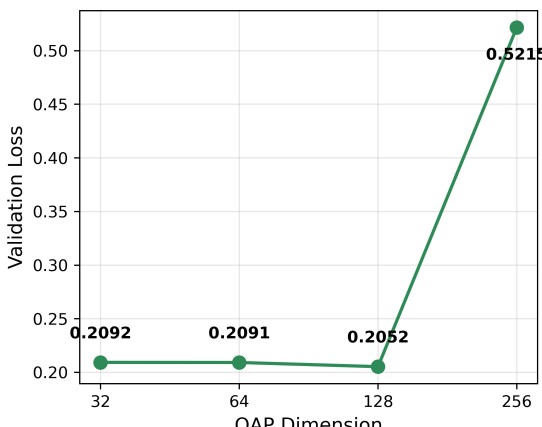

Figure 5: Validation loss across QAP dimensions using the iTransformer backbone. the best result is obtained at $d_{\text{qap}} = 128$.

Table 7: Comparison of forecasting performance with $F_{\max} = 20$ vs $F_{\max} = 30$ on the Traffic dataset.

| Model | $F_{\max} = 20$ | | $F_{\max} = 30$ | |
|---|---|---|---|---|
| | MSE | MAE | MSE | MAE |
| iTransformer | 0.3581 | 0.3317 | **0.1502** | **0.2057** |
| TimeMixer | 0.3564 | 0.3328 | **0.1701** | **0.2208** |
| DLinear | 0.3719 | 0.3484 | **0.1999** | **0.2390** |
| SpikeRNN | 0.3794 | 0.3392 | **0.1847** | **0.2298** |
| Spikformer | 0.3845 | 0.3560 | **0.1790** | **0.2449** |

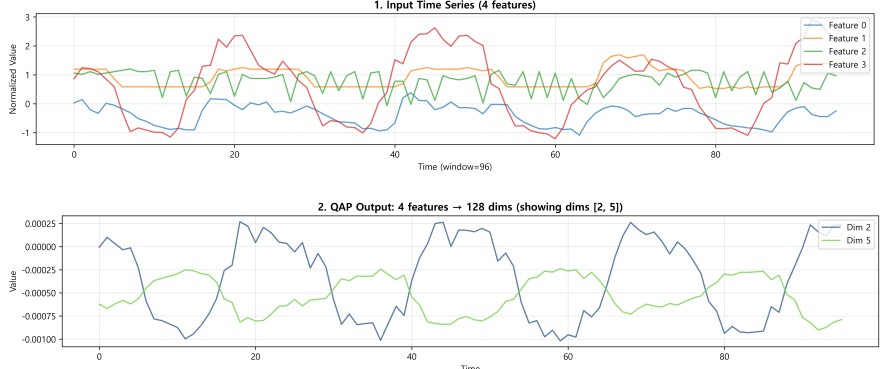

Figure 6: Visualization of QAP transformation on the first client from the Electricity dataset. (Top) Raw input time series with 4 features. (Bottom) Two representative dimensions from the 128-dimensional QAP output.

## A.5 Limitations and Future Work

While FedQAPer demonstrates promising results under feature-level heterogeneity, several limitations remain. First, in the extreme case where a client has only a single feature channel ($F_i = 1$), the attention mechanism of QAP degenerates into a trivial self-attention mapping, reducing its benefit to that of a linear projection. Second, due to the FedPer design that applies personalized heads at the client side, backbone architectures must exclude intrinsic heads to avoid over-parameterization and performance degradation, which restricts the direct use of certain state-of-the-art forecasting backbones.

As future work, we plan to investigate enhanced QAP variants that can maintain expressiveness even when $F_i = 1$, for instance by incorporating cross-client regularization or slot-level contrastive objectives. In addition, we aim to develop a more flexible backbone–head decoupling mechanism that enables the reuse of head-equipped architectures without redundancy, possibly through selective head freezing or shared head distillation. These directions would further improve the robustness and generality of FedQAPer in diverse practical federated scenarios.

