# OpenReview forum: "FedQAPer: Query Attention Pooling for Dimension Alignment in Federated Non-IID Time-series Forecasting with Personalized Heads"
_ICLR.cc/2026/Conference — ICLR 2026 Conference Desk Rejected Submission_

### Official Review · Reviewer_gxN3 · 2025-10-25

**Soundness:** 2
**Presentation:** 2
**Contribution:** 3
**Rating:** 2
**Confidence:** 4

**Summary:**

This work proposes a federated time series learning framework FedQAPer to solve the problem of feature-level non-independent and identically distributed (non-IID). QAP projects heterogeneous features into a unified latent space, enabling collaborative training of the shared backbone with different input dimensions. At the same time, FedPer uses a personalized head to preserve the client's personalized capabilities. Extensive experimental results on several datasets show that FedQAPer outperforms existing methods.

**Strengths:**

1. Overall, this paper is well-written and easy to follow.

2. This work introduces feature-level non-IID for federated learning of time series is novel.

3. Extensive experimental results are promising.

**Weaknesses:**

1. The introduction section does not explain the feature clearly. Can it be understood as a variate in the context of time series? The main contribution lies in the challenge of different dimensional features, but a lack of the combination of QAP and FedPer.

2. QAP maps any $F_i$ to a fixed $d_{qap}$, and then the personalized head maps it back to $F_i.$ The paper does not discuss whether this mapping can preserve the cross-information between the original features.

3. The baselines are compared with time series models rather than the FL for the time-series framework.

4. The experiments in this work consider the participation of all clients, but real-world scenarios usually involve partial participation, which should also be verified.

5. Only parts of the results in Table 2 have confidence intervals. It would be better to provide confidence intervals for both Tables 1 and 2.

6. There are many commonly considered datasets for time series, but the experiments only test two, which are not solid enough.

7. The work mentions that SNN has advantages in energy consumption/communication, but the article does not consider the computational and communication overhead.

8. The presentation of figures could be improved. The font in Figure 1 is too small, and Figure 1 could be further improved. While Figure 2 and 3 are relatively big.

**Questions:**

Shown in weaknesses.

---

> ### Author Response · Authors · 2025-11-21
> **Rebuttals (1/2)**
>
> We sincerely thank the reviewer for the thoughtful feedback and for recognizing the novelty of addressing feature-level non-IID in federated time-series learning. We appreciate the positive assessment of our experimental results and presentation clarity.
> We have addressed each of your concerns as follows.
>
> > **1. The introduction section does not explain the feature clearly. Can it be understood as a variate in the context of time series? The main contribution lies in the challenge of different dimensional features, but a lack of the combination of QAP and FedPer.**
>
> **Response:**
> Thank you for pointing this out. Yes, "feature" refers to a variate in the time-series context—we will clarify this terminology in the revised introduction. Regarding the contribution, we acknowledge that the synergy between QAP and FedPer was not sufficiently emphasized. We will enhance Figure 1 and the corresponding explanation to clearly show how QAP enables feature alignment across heterogeneous clients while FedPer preserves client-specific personalization.
>
>
> > **2. The paper does not discuss whether this mapping can preserve the cross-information between the original features.**
>
> **Response:**
> Thank you for raising this concern. However, our design ensures that cross-feature information is effectively preserved through complementary mechanisms:
>
> 1. **Information Expansion (Scalar $\to$ Vector):**
> It is important to clarify that QAP does not simply compress raw scalar features. Instead, it first **projects** each raw scalar value into a high-dimensional vector ($1 \to d_{qap}$). For example, if a client has 10 features and $d_{qap}=128$, the input information is expanded from 10 scalars to 1,280 latent values ($10 \times 128$) before aggregation. This high-dimensional embedding space provides sufficient capacity to encode both the value and the identity (via slot embeddings) of each feature.
>
> 2. **Side-Channel Fusion (Complementing with Global Statistics):**
> While the attention mechanism with $Q=1$ acts as a **selective filter** (focusing on the most salient features), it might miss global correlations. To resolve this, we incorporated **Side-Channel Fusion** which concatenates statistical summaries (Mean-pooling and Max-pooling) with the attention output.
> * **Attention:** Captures non-linear, weighted cross-feature relationships.
> * **Mean-Pooling:** Preserves the global trend and background information of all features.
> * **Max-Pooling:** Captures critical peak signals or anomalies across features.
>
> By fusing these three representations, the final latent vector $z_t$ encapsulates both the detailed relationships and the holistic statistics of the original features, thus preserving cross-feature information through this complementary design.
>
>
> > **3. The baselines are compared with time series models rather than the FL for the time-series framework.**
>
> **Response:**
> We wish to clarify that a direct comparison with standard FL frameworks (e.g., FedAvg, FedProx) is **structurally infeasible** in our setting. These methods strictly assume that all clients share identical feature spaces ($F_i = F_j$). This assumption fundamentally conflicts with our problem definition, where each client possesses heterogeneous feature dimensions.
>
> Without an alignment mechanism like QAP, these baselines cannot even be initialized, as there is no mathematical way to directly aggregate model parameters (specifically the input layers) with mismatched dimensions. This is precisely the technical gap FedQAPer addresses.
>
> Therefore, we adopted the following comparison strategy:
> * **Local-only training:** Shows the value of federated learning over isolated training
> * **Projection baseline (added in revised paper):** Validates the superiority of QAP over naive alignment
>
> These baselines directly evaluate the benefits of our approach under feature heterogeneity.
>
>
> > **4. The experiments in this work consider the participation of all clients, but real-world scenarios usually involve partial participation, which should also be verified.**
>
> **Response:**
> We agree that partial participation is a critical factor in real-world FL scenarios. While we focused on full participation in the initial submission to establish a performance upper bound, we are currently conducting additional experiments with **partial client participation**. We will include these results in the revised version to demonstrate the robustness of our framework.
>
>
> > **5. Only parts of the results in Table 2 have confidence intervals. It would be better to provide confidence intervals for both Tables 1 and 2.**
>
> **Response:**
> Thank you for pointing this out. We will add confidence intervals to Table 1 for consistency.

---

> ### Author Response · Authors · 2025-11-21
> **Rebuttals (2/2)**
>
> > **6. There are many commonly considered datasets for time series, but the experiments only test two, which are not solid enough.**
>
> **Response:**
> Response:
> We chose Traffic and Electricity specifically because they have **high feature dimensionality (F>300)**, which is essential for our experimental setup. To simulate realistic feature heterogeneity across 20+ clients, we need a large feature pool to partition.
>
> Most common time-series benchmarks (e.g., ETT, Exchange, Weather) have fewer than 10 features, making them unsuitable for meaningfully testing feature-level non-IID scenarios. Without sufficient features, we cannot create the diverse client partitions our method is designed to handle.
>
>
> > **7. The work mentions that SNN has advantages in energy consumption/communication, but the article does not consider the computational and communication overhead.**
>
> **Response:**
> Thank you for raising this point. We acknowledge that our current manuscript lacks quantitative analysis of computational efficiency. In the revision, we will add a comparative analysis of **inference energy consumption** between ANN-based (iTransformer) and SNN-based (iSpikeformer) backbones to demonstrate the efficiency benefits of SNNs within FedQAPer.
>
> > **8. The presentation of figures could be improved. The font in Figure 1 is too small, and Figure 1 could be further improved. While Figure 2 and 3 are relatively big.**
>
> **Response:**
> Thank you for the constructive feedback regarding the visualization.We will increase the font size in Figure 1 and rescale Figures 2 and 3 for better visual consistency in the revision. The updated figures will be included in the revised paper.

---

### Official Review · Reviewer_QtAY · 2025-10-31

**Soundness:** 1
**Presentation:** 1
**Contribution:** 1
**Rating:** 2
**Confidence:** 5

**Summary:**

This paper proposes FedQAPer, a personalized federated learning framework for heterogeneous time series forecasting. FedQAPer is built on a scenario where each client holds time series data with different variables. Based on the FedPer framework, the authors introduce a Query Attention Pooling (QAP) module within FedQAPer to align heterogeneous client features into a unified latent space, and incorporate a personalized head on each client without communication to achieve personalization. In addition, the authors conduct experiments using five time series forecasting backbones on two multi-point time series datasets.

**Strengths:**

**S1.** The authors provide complete source code for reproduce.

**S2.** The challenge of feature heterogeneity in real-world time series forecasting is make sense.

**Weaknesses:**

**W1.** Line 63, the authors claim their work 'unlike prior work' and 'constrcut a more realistic FL setting where each client has a different number and type of feature'. However, the environment that each client has a different number of type and type of feature has been well-explored in prior work like [1, 2].

[1] Federated foundation models on heterogeneous time series. AAAI 2025.

[2] Tackling data heterogeneity in federated time series forecasting. arXiv 2024.

**W2.** Regarding the author's claim on Line 102, distributing from a single dataset into clients carrying varying numbers of features is less challenging than ***W1*** because they are to some extent homogeneous. Therefore, the term “limited research” here does not imply that the topic was unexplored due to its difficulty, but rather that further exploration is unnecessary.

**W3.** Section 2.2 is ambiguous. There is a wrong citation in Line 127 (i.e., FedLECYu et al. (2025)).

**W4.** The notation in Algorithm 1 and Algorithm 2 is confusing and difficult to understand.

**W5.** FedQAPer is a fully incremental effort, largely built upon existing methods such as FedPer's personalization mechanism and existing deep forecasting models. The author's description of FedQAPer in Section 3 inevitably leans heavily toward engineering details, resembling more of a course assignment.

**W6.** The experimental section is too weak and insufficient:

* Only two datasets were considered, and the configuration was not optimal. Since each variable in Traffic and Electricity represents data from a single site, the authors' partitioning strategy (i.e., assigning different numbers of sites to a client to simulate heterogeneity) should have prompted them to first understand the dataset characteristics.
* It is unclear what the purpose of Table 1 is, as it only presents a comparison of various existing deep learning models within the FedQAPer framework, while the section highlighting FedQAPer's effectiveness (superiority) is absent (i.e., comparison with other FL/PFL algorithms).
* Framework analysis and hyperparameter analysis are missing.

**Questions:**

Please see Weaknesses.

---

> ### Author Response · Authors · 2025-11-21
> **Rebuttals (1/2)**
>
> We sincerely thank the reviewer for their time and valuable feedback. We are encouraged that the reviewer recognized the feature heterogeneity challenge in real-world time series forecasting as a meaningful and practical problem. We also appreciate the acknowledgment of our efforts toward **reproducibility by providing the complete source code.
>
> We have carefully reviewed the concerns raised regarding the soundness and contribution of our work. In the response below, we have provided detailed clarifications and additional context to better explain our methodology and experimental results. We hope these responses help address the issues you identified.
>
> > **1. The reviewer questions the novelty of our claim regarding the "realistic FL setting where each client has a different number and type of feature," citing prior works (e.g., "Federated foundation models on heterogeneous time series") that appear to have explored similar environments.**
>
> **Response:**
> We respectfully wish to clarify the distinction between our work and the cited references.
> The studies mentioned ([1] Federated foundation models on heterogeneous time series. AAAI 2025. and [2] Tackling data heterogeneity in federated time series forecasting. arXiv 2024.) primarily address **data type** or **domain** heterogeneity (e.g., different clients holding entirely different datasets or distributions). They do not address the specific structural challenge where clients within the same domain hold **varying numbers of features** ($F_i \neq F_j$).
> Our work specifically targets this dimensional misalignment, which prevents the direct application of standard FL frameworks. FedQAPer is designed to resolve this specific structural constraint, which is distinct from the problems tackled in [1] and [2].
>
> > **2. Regarding the author's claim on Line 102, distributing from a single dataset into clients carrying varying numbers of features is less challenging than W1 because they are to some extent homogeneous. Therefore, the term “limited research” here does not imply that the topic was unexplored due to its difficulty, but rather that further exploration is unnecessary.**
>
> **Response:**
> We acknowledge the reviewer's perspective, but we strongly believe this setting is practically vital.
> Consider a real-world **Smart Grid** scenario: A client represents a distribution panel, and its features correspond to sub-meters (sub-loads). Even if we group sub-loads by energy type (making them somewhat homogeneous), different panels inevitably manage different numbers of loads (e.g., Panel A monitors 5 HVAC units, Panel B monitors 2).
> In this case, standard FL fails because the input layers do not match. FedQAPer handles this by using **Slot Embeddings** to capture the semantic identity of each sub-load (e.g., "HVAC" vs. "Lighting") regardless of the count. Therefore, we argue that this research is necessary to enable FL in such realistic, hardware-constrained environments.
>
> > **3. Section 2.2 is ambiguous. There is a wrong citation in Line 127 (i.e., FedLECYu et al. (2025)).**
>
> **Response:**
> We apologize for the oversight. We will correct the citation and clarify the ambiguity in Section 2.2 in the revised paper.
>
> > **4. The notation in Algorithm 1 and Algorithm 2 is confusing and difficult to understand.**
>
> **Response:**
> Thank you for pointing this out. We will thoroughly revise the mathematical notations in Algorithms 1 and 2 to ensure they are consistent and easy to follow in the  revised paper.
>
> > **5. FedQAPer is a fully incremental effort, largely built upon existing methods such as FedPer's personalization mechanism and existing deep forecasting models. The author's description of FedQAPer in Section 3 inevitably leans heavily toward engineering details, resembling more of a course assignment.**
>
> **Response:**
> We acknowledge the reviewer's point. However, we note that most existing FL studies assume clients have identical feature counts ($F_i = F_j$), , which does not reflect real-world sensor heterogeneity. FedQAPer was designed specifically to address this gap and enable federated time-series forecasting under heterogeneous feature spaces. Our contribution lies in this architectural adaptation that makes impossible FL scenarios feasible.

---

> ### Author Response · Authors · 2025-11-21
> **Rebuttals (2/2)**
>
> > **6. The experimental section is too weak and insufficient (Datasets, Table 1 Purpose, Hyperparameter analysis).**
>
> **Response:**
> * **Datasets:** We selected **Traffic** and **Electricity** specifically because they possess high dimensionality ($F > 300$), which allows us to simulate diverse and realistic client partitions ($F_i$ ranging from 1 to $N$). Most other public time-series datasets have too few features to meaningfully simulate this setting.
> * **Table 1 Purpose:** The goal of Table 1 is to demonstrate **compatibility with various backbones**. Since standard FL requires identical architectures, showing that FedQAPer works across diverse ANNs and SNNs confirms the framework's generalizability.
> * **Comparisons:** As explained in responses to other reviewers, comparing with other FL algorithms (e.g., FedAvg) is structurally impossible without an alignment module like ours. Instead, we focused on comparing against **Local-only** (to show FL benefits) and a **Projection Baseline** (will be added in the revised paper to show QAP benefits) to validate effectiveness.

---

> > ### Comment · Reviewer_QtAY · 2025-11-26
> >
> > Thanks for the response. After carefully reviewing the response, I still maintain my original position for the following reasons:
> >
> > **For W1, W2**, the proposed FedQAPer focuses on dimensional misalignment ***within the same domain*** (clients possess different numbers of features). This is a lower-level and comparatively easier problem than the settings in references [1,2], where clients hold ***fundamentally different cross-domain time series with inherently different dimensionalities***. Therefore I still do not clearly see the motivation or practical significance of the problem addressed in the paper.
> >
> > **For W5**, based on the concerns raised in W1 and W2, the claim that “enabling federated time-series forecasting under heterogeneous feature spaces is meaningful” does not hold convincingly.
> >
> > **For my W6**, evaluating only two standard TSF datasets is insufficient. More complex, high-dimensional datasets such as Solar (5162-dimension), M5 (3049-dimension), and Wiki-20k (20000-dimension) could be consider to included. In addition, comparing against personalized FL baselines is essential. ***Simply replacing models (like different ANN/SNN) within the same framework does not provide meaningful insight, especially for a paper that proposes a personalized FL method***.
> >
> > I encourage the authors to reconsider the core motivation and the practical significance of the problem addressed in this work.

---

> ### Author Response · Authors · 2025-11-30
> **Regarding W1, W2, and W5 (Practical Motivation) and W6 (Datasets)**
>
> Thank you for your thoughtful follow-up and for taking the time to carefully reconsider our response. We appreciate your constructive feedback, and we acknowledge that the issues you raised are indeed important aspects of heterogeneity in federated learning.
>
> **Regarding W1, W2, and W5 (Practical Motivation):**
>
> We respectfully argue that **same-domain dimensional misalignment is also a practically significant problem**. Our research originated from a **real-world application**, forecasting sub-load consumption in electrical distribution panels of non-residential buildings.
>
> In this scenario, non-residential buildings generally share similar overall consumption patterns due to clear distinctions between working and non-working hours. Since all data are based on electrical power consumption, this is not a cross-domain setting. However, the **number of sub-loads varies across different distribution panels**, making it hard to use same backbone model due to this misalignment. This **dimensional misalignment is a concrete and practical challenge**.
>
> Furthermore, while existing research has addressed heterogeneity in terms of dataset types or data distributions, there is **limited work that specifically tackles feature-level heterogeneity** across clients. We believe this gap further justifies the significance of our research.
>
> We agree that cross-domain scenarios, as addressed in [1,2], are important. However, we believe that the dimensional misalignment problem within the same domain also deserves attention, as it arises naturally in real-world deployments.
>
> **Regarding W6 (Datasets):**
>
> Thank you for suggesting additional datasets such as Solar, M5, and Wiki-20k. We attempted to locate these datasets but encountered some confusion—the dataset names and dimensionalities you mentioned do not fully match what we found (e.g., on Kaggle). Could you kindly clarify the specific sources or references for these datasets?
>
> We appreciate this valuable suggestion, and if we can identify the correct datasets, we will conduct additional experiments in future work.
>
> Thank you again for your time and constructive comments.

---

### Official Review · Reviewer_PcMa · 2025-10-31

**Soundness:** 2
**Presentation:** 2
**Contribution:** 2
**Rating:** 2
**Confidence:** 4

**Summary:**

The paper presents FedQAPer, a federated learning framework aimed at addressing feature heterogeneity among clients in time-series forecasting. It introduces a Query Attention Pooling module that maps each client’s local features into a shared latent space, enabling collaborative training of a common model while maintaining client-specific personalization heads, similar to FedPer. The framework is evaluated on the Electricity and Traffic datasets using multiple backbone architectures, including iTransformer, TimeMixer, DLinear, SpikeRNN, and Spikeformer. The results indicate that FedQAPer enhances both global and personalized model performance across all tested configurations.

**Strengths:**

Effectively tackles a key challenge in federated learning: managing clients with heterogeneous feature spaces, an aspect often overlooked in previous studies.

The proposed approach is simple and adaptable, making it easily integrable with various backbone architectures.

**Weaknesses:**

•The method mainly combines existing ideas like feature projection and personalized heads. It feels close to FedPer/FedProx with an added pooling layer. The authors do not clearly explain what new insight QAP offers beyond a simple shared linear projection.

•The paper doesn’t compare against straightforward baselines such as zero-padding, global MLP projection, or other feature-alignment FL methods. This makes it unclear whether QAP itself brings any real benefit.

•Unclear aggregation of metrics: its written in the paper that “Each client predicts a subset of features,” but still reports a single global MAE and MSE across all clients. It’s not explained how these are computed fairly when clients have different targets.

•In Table 5, when Fmax increases from 20 to 30, the iTransformer MAE suddenly drops from 0.3317 to 0.2057 — a huge jump that seems unrealistic. This raises a concern about a possible setup or data leakage issue.

•SNN claims not supported by experiments. The paper states that “Spike-based architectures in FedQAPer yield energy-efficient communication and faster inference” (Section 4.4), but no energy or latency results are provided. Without such experiments, these claims are not validated.

•The method involves transmitting shared parameters from QAP, but the paper doesn’t analyze communication overhead or possible privacy leakage.

•The proposed model feels like an extension of FedPer with attention-based pooling. It lacks deeper analysis or strong theoretical justification.

**Questions:**

How is QAP fundamentally different from using a shared linear layer or MLP to project each client’s features into a fixed dimension?

Did you test with partial client participation instead of all clients joining every round?

How exactly do you compute the global MAE/MSE when clients have different output features?

Why does performance change so drastically in Table 5 when Fmax increases from 20 to 30? Could this be due to different data splits or leakage?

Did you run experiments with multiple random seeds or client partitions to ensure stability?

What is the computational cost of QAP on clients compared to FedPer without QAP?

Does sharing QAP parameters risk leaking feature-level information from clients? Any privacy considerations?

---

> ### Author Response · Authors · 2025-11-21
> **Rebuttals (1/2)**
>
> We thank the reviewer for the insightful technical questions, particularly regarding the mechanism of QAP and its privacy implications. We have addressed each point below.
>
> > **1. How is QAP fundamentally different from using a shared linear layer or MLP to project each client’s features into a fixed dimension?**
>
> **Response:**
> There is a fundamental difference in how they handle feature heterogeneity. A simple shared linear layer or MLP would merely force the output dimensions to match numerically, without accounting for the differing semantic meanings or varying quantities of input features.
>
> In contrast, QAP does not just force feature dimensions to match numerically. It actively **aligns** heterogeneous feature spaces into a unified representation through a learnable three-stage process:
>
> * **Value Projection (Shared):** First, a shared linear layer projects raw scalar values into high-dimensional vectors. This allows the model to process "values" consistently across all clients.
> * **Feature Embedding (Local):** Crucially, since Client A’s "Feature 1" might be different from Client B’s "Feature 1", we add a learnable, client-specific Feature Embedding. This injects the semantic identity of each feature locally.
> * **Query Attention (Shared):** Finally, a shared Query vector (learned by the server) attends to these feature-embedded representations. This mechanism effectively "queries" and aggregates relevant information from the variable-length feature sets of different clients into a fixed latent space.
>
> Unlike a simple projection, this process ensures that the backbone receives a representation that is not just dimensionally consistent, but **semantically aligned**, enabling effective collaborative learning despite feature disparities.
>
> Specifically, we employed the **iSpikeformer**—a newly introduced backbone in our framework—on the Traffic dataset (batch size: 128, window: 96). The comparison below clearly demonstrates that QAP outperforms the projection baseline, proving its superior capability in feature alignment.
>
> | **Horizon** | **MSE (Projection)** | **MSE (QAP)** | **MAE (Projection)** | **MAE (QAP)** |
> | :---: | :---: | :---: | :---: | :---: |
> | 192 | 1.1403 |**0.7109** | 0.7604 |**0.5433**|
> | 96 | 0.9216 |**0.5663**| 0.6704 |**0.4574**|
> | 48 | 1.0053 |**0.4832**| 0.7045 |**0.4172**|
>
> > **2. Did you test with partial client participation instead of all clients joining every round?**
>
> **Response:**
> In the submitted manuscript, we focused on the full participation setting to establish a baseline. This study is started from forecasting distribution panels and all distribution panels should participate the federated learning for better result. But we also acknowledge that partial client participation is also important in other scenarios. Thank you for pointing the problem and we will conduct extra case study if possible.
>
> > **3. How exactly do you compute the global MAE/MSE when clients have different output features?**
>
> **Response:**
> Since clients possess heterogeneous output feature dimensions (i.e., $C_i \neq C_j$), direct aggregation or concatenation of raw prediction tensors is structurally impossible. Therefore, our implementation computes global metrics using **dataset-size weighted averaging** of locally computed scalar metrics. The detailed process is as follows:
>
> 1. **Local Metric Computation:**
> First, each client $i$ calculates the scalar metric (e.g., MAE) on its own local feature subset using its personalized head output. This is averaged across all dimensions (Batch $B$, Horizon $H$, and Features $C_i$) within the client:
>
> $$
> \text{MAE}\_i = \frac{1}{B \cdot H \cdot C\_i} \sum\_{b,h,c} |y\_{\text{true}}^{(i)} - y\_{\text{pred}}^{(i)}|
> $$
>
> where $C_i$ is the unique feature count for client $i$, and $y^{(i)}$ represents the tensor of shape $[B, H, C_i]$.
>
> 2. **Global Weighted Aggregation:**
> The server then aggregates these scalar metrics. To ensure fair evaluation, each client's contribution is weighted by the size of its local training dataset $|\mathcal{D}_i|$:
>
> $$
> \text{MAE}\_{\text{global}} = \sum\_{i=1}^{N} \left( \text{MAE}\_i \times \frac{|\mathcal{D}\_i|}{\sum\_{j=1}^{N} |\mathcal{D}\_j|} \right)
> $$
>
> This approach allows us to evaluate global performance while preserving the local context of each client's unique feature space. We also note that all metrics are computed on a normalized scale (using a StandardScaler fitted on the training split) to maintain consistency across clients with different value ranges.

---

> ### Author Response · Authors · 2025-11-21
> **Rebuttals (2/2)**
>
> > **4. Why does performance change so drastically in Table 5 when Fmax increases from 20 to 30? Could this be due to different data splits or leakage?**
>
> **Response:**
> The performance improvement is not due to data leakage (validation/test sets are strictly separated), but rather due to **increased feature coverage and overlap**.
>
> With $N=20$ clients and a maximum feature count of $F_{max}=30$, the total number of feature slots across all clients reaches up to 600 ($20 \times 30$). This represents a 50% increase in capacity compared to the $F_{max}=20$ setting (which sums to 400 features). Sampling up to 600 features increases feature overlap among clients and covers more of the total feature space. This overlap allows more clients to observe critical variables simultaneously, enabling the shared backbone to learn stronger global dependencies.
>
> > **5. Did you run experiments with multiple random seeds or client partitions to ensure stability?**
>
> **Response:**
> We primarily utilized Seed 42 for the reported results. However, the fact that FedQAPer consistently outperforms baselines across **multiple different backbone architectures** (ANNs and SNNs) and **two distinct datasets** serves as strong evidence that the performance is not a result of random chance.
>
>
> > **6. What is the computational cost of QAP on clients compared to FedPer without QAP?**
>
> **Response:**
> **FedPer without QAP** cannot exist in our setting—the backbone requires fixed-dimension inputs and cannot handle varying ($F_i$) across clients. QAP is essential to enable federated learning under feature heterogeneity.
> Regarding computational cost, QAP is lightweight compared to the backbone. We will add a parameter count and FLOPs analysis in the revision to quantify the overhead(e.g., QAP comprises only 23% of iSpikeformer's parameters).
>
>
> > **7. Does sharing QAP parameters risk leaking feature-level information from clients? Any privacy considerations?**
>
> **Response:**
> Thank you for this important question. QAP parameters are shared with the server, but they are **generic transformation matrices** (query, value projection, attention operations) that work on any feature set without encoding feature-specific information.
> The critical feature identities are protected through slot embeddings, which remain strictly local and are never transmitted. Without these embeddings, the server cannot determine which specific features each client possesses.

---

### Official Review · Reviewer_5jVQ · 2025-10-31

**Soundness:** 3
**Presentation:** 3
**Contribution:** 2
**Rating:** 4
**Confidence:** 3

**Summary:**

This paper introduces FedQAPer, a novel federated learning framework for time-series forecasting designed to handle feature heterogeneity, where clients have different input features. The method uses Query Attention Pooling (QAP) to project heterogeneous client features into a unified space, enabling collaborative training of a shared backbone model. It combines this with the FedPer algorithm, which uses personalised heads on each client to map the unified representations back to their original feature dimensions, capturing local patterns. Experiments show FedQAPer is compatible with both ANN and SNN backbones and effectively improves forecasting performance.

**Strengths:**

1. The feature-level non-IID problem is a core challenge in real-world FL deployments (especially involving heterogeneous sensor networks). Research on this specific problem is still relatively limited, making the paper's focus highly valuable.
2. The solution proposed by FedQAPer is intuitive and sound. Using a learnable module (QAP) to achieve dimensional alignment, combined with FedPer for personalisation, is a logical path to solving this problem.
3. he design of QAP considers robustness. In addition to cross-attention, it introduces a "side-channel fusion" of statistical features (mean, max). This is a good design choice to prevent the attention mechanism from becoming unstable or degenerating when the number of features is very small.

**Weaknesses:**

1. In the ablation study (Table 2), the authors compare FedPer (i.e., FedQAPer) against "Local-only" (clients trained independently). The authors claim FedPer "consistently outperforms" local-only training. However, the data in Table 2 clearly show that for the Electricity dataset, the "Local" versions of both TimeMixer and Spikeformer performed better (lower error) than the "Fed" versions on both MSE and MAE metrics.
TimeMixer (Electricity): Fed (MSE 0.220, MAE 0.325) vs Local (MSE 0.165, MAE 0.276).
Spikeformer (Electricity): Fed (MSE 0.232, MAE 0.339) vs Local (MSE 0.228, MAE 0.325).
This indicates that, in these cases, the federated collaboration (knowledge sharing) actually led to negative transfer, harming model performance. This finding severely undermines the paper's core argument that FedQAPer provides a clear benefit over isolated training. The authors must provide a deep analysis and explanation for this phenomenon.
2. The main results (Table 1) only show the performance of different backbones under the FedQAPer framework. It does not compare FedQAPer as a holistic method against other federated learning algorithms.

**Questions:**

See weakness.

---

> ### Author Response · Authors · 2025-11-21
> **Rebuttal**
>
> We genuinely appreciate the detailed review of our work and the positive assessment of our contributions. We are pleased that the reviewer recognized the **feature-level non-IID problem** as a core challenge in real-world deployments and found our focus on this under-researched area highly valuable. We are also encouraged that the reviewer considered our proposed FedQAPer framework—combining QAP for alignment and FedPer for personalization—to be an intuitive and sound solution. Furthermore, we appreciate the recognition of our robust design choices, specifically the "side-channel fusion" mechanism that prevents attention instability. We are grateful for the high scores on soundness and presentation.
>
> Below, we respond to specific comments to clarify our contributions and the trade-offs involved.
> > **1. In the ablation study (Table 2), the reviewer notes that for the Electricity dataset (TimeMixer, Spikeformer), "Local-only" training outperformed FedQAPer, suggesting negative transfer. The reviewer asks for a deep analysis of this phenomenon.**
>
> **Response:**
> We thank the reviewer for this critical observation. We acknowledge that in the specific cases of TimeMixer and Spikeformer on the Electricity dataset, the **Local-only** approach yielded lower errors than the Federated approach. We agree that this suggests a distinct occurrence of negative transfer, where the global backbone—generalized across diverse clients—may have diluted strong client-specific temporal patterns present in the Electricity dataset.
>
> However, we would like to highlight two critical perspectives that justify the value of FedQAPer despite these specific cases:
>
> * **Efficiency and Scalability:** While "Local-only" training can sometimes achieve high accuracy by overfitting to local data, it is computationally expensive and unscalable. In a real-world IoT scenario, training full-sized complex models (like Transformers) independently on edge devices consumes significant local resources and time. FedQAPer offloads representation learning to the shared backbone, reducing local computation.
> * **Privacy-Utility Trade-off:** The primary goal of FL is to enable model training **without data centralization**. Even if FL shows slightly lower performance than local training in specific outlier cases due to heterogeneity, it remains the only viable solution when data privacy regulations prevent centralized aggregation.
>
> In the revision, we will acknowledge these cases and discuss the inherent privacy-efficiency-accuracy trade-off in federated settings.
>
> > **2. The main results (Table 1) only show the performance of different backbones under the FedQAPer framework. It does not compare FedQAPer as a holistic method against other federated learning algorithms.**
>
> **Response:**
> The primary reason for not comparing against standard FL algorithms (e.g., FedAvg, FedProx) is the **structural constraint** imposed by feature heterogeneity. Standard algorithms require identical model architectures across all clients to aggregate parameters ($\theta_{global} \leftarrow \sum w_i \theta_i$). In our setting, where each client has a different input dimension $F_i$, the input layers and consequent structures are inherently mismatched, making direct aggregation impossible without a personalization strategy like FedPer.
>
> However, to address the reviewer’s concern, we analyzed the potential cost of a **"FedAvg"** aggregation where personalized heads are transmitted. If we assume a client has 10 features, the parameter count for the prediction head alone reaches approximately **614,880**. If all clients were to transmit these head parameters to the server for aggregation, it would result in a massive surge in communication overhead.
>
> This demonstrates that FedQAPer (where heads are kept local) is not just an architectural choice but a necessity for communication efficiency in feature-heterogeneous settings. We will include the detailed calculation process and a comprehensive cost analysis in the revised version of the paper.

---

### Official Review · Reviewer_wiXn · 2025-11-01

**Soundness:** 1
**Presentation:** 3
**Contribution:** 2
**Rating:** 2
**Confidence:** 4

**Summary:**

This paper proposes FedQAPer, a federated learning framework for non-IID time-series forecasting where clients have heterogeneous feature dimensions. It extends FedPer by adding a Query Attention Pooling (QAP) module that projects each client's features into a shared latent space for dimension alignment before federated aggregation, while keeping personalized output heads. The method is evaluated under different ANN (iTransformer, TimeMixer, DLinear) and SNN (SpikeRNN, Spikeformer) backbones on the Electricity and Traffic datasets. However, the experiments appear to present only FedQAPer's own results, with no baseline comparisons or ablations, making it challenging to assess relative performance or the actual benefit of QAP.

**Strengths:**

The work addresses a relatively understudied challenge in federated learning: heterogeneity in feature dimensions across clients, which is distinct from the usual label or distributional non-IID settings. The proposed Query Attention Pooling (QAP) mechanism is a conceptually simple and general idea that could, in principle, be applied to other settings as well. The paper is well written and organized, with clear diagrams explaining the role of QAP and personalized heads in the pipeline.

**Weaknesses:**

The paper reports only FedQAPer's own results across backbones, with no quantitative comparisons against established federated baselines, making it challenging to judge its relative performance.

There is no evaluation isolating the effect of the Query Attention Pooling (QAP) module, so its actual contribution is unclear.

The method is a somewhat incremental architectural extension of FedPer, reusing its shared-backbone and personalized-head structure with a minor attention-based preprocessing layer.

The paper provides no formal/theoretical reasoning showing that QAP achieves meaningful feature alignment across clients.

**Questions:**

How does FedQAPer quantitatively differ from FedPer? Can you show a side-by-side comparison or ablation to isolate the impact of the Query Attention Pooling?

Why are no baseline results included? Without benchmarking, how should readers interpret the claimed improvements?

How is feature heterogeneity simulated? Are the random feature splits representative of real-world multi-sensor or multimodal setups?

Can you provide any empirical evidence of alignment (e.g., embedding similarity or visualization) to confirm that QAP actually harmonizes client feature spaces?

What is the communication and computational overhead introduced by QAP compared to FedPer?

Does the framework generalize to partial feature overlap or clients with entirely disjoint features?

Could QAP be integrated into other personalized FL schemes or is it somehow specific to FedPer-style architectures?

Have you considered any real-world non-synthetic datasets where feature heterogeneity naturally occurs (e.g., multi-site sensor networks or IoT data)?

Why does the validation loss across QAP dimensions using the iTransformer backbone jumps so suddenly for d_QAP = 256?

---

> ### Author Response · Authors · 2025-11-21
> **Rebuttals (1/2)**
>
> We sincerely thank the reviewer for their constructive feedback and for recognizing the value of our work. We are encouraged that the reviewer appreciates our focus on feature dimension heterogeneity, identifying it as a distinct and understudied challenge in federated learning. We are also pleased that the reviewer found our proposed Query Attention Pooling (QAP) mechanism to be a conceptually simple and general idea with potential applicability to other settings. Furthermore, we appreciate the positive remarks regarding the clarity of our presentation and diagrams.
>
> Below, we carefully address the concerns regarding baseline comparisons and ablations to clarify the relative performance and benefits of FedQAPer.
>
> > **1. How does FedQAPer quantitatively differ from FedPer? Can you show a side-by-side comparison or ablation to isolate the impact of the Query Attention Pooling?**
>
> **Response:**
> We appreciate this question. The fundamental difference lies in the capability to handle heterogeneous feature spaces. The standard FedPer framework requires all clients to have an identical input dimension to share the backbone model. Therefore, it is structurally impossible to apply vanilla FedPer directly to our setting where clients have different numbers of features.
>
> FedQAPer solves this by introducing **Query Attention Pooling (QAP)**, which aligns diverse input dimensions into a unified latent dimension, enabling the sharing of the backbone.
>
> To isolate the impact of QAP, we compared it against a baseline employing a simple **linear projection** where client features are mapped to the dimension $d_{qap}$ via distinct linear layers. The results below present the evaluation on the test dataset using the trained model weights. Specifically, we employed the **iSpikeformer**—a newly introduced backbone in our framework—on the Traffic dataset (batch size: 128, window: 96). The comparison clearly demonstrates that QAP outperforms the projection baseline, proving its superior capability in feature alignment.
> | **Horizon** | **MSE (Projection)** | **MSE (QAP)** | **MAE (Projection)** | **MAE (QAP)** |
> | :---: | :---: | :---: | :---: | :---: |
> | 192 | 1.1403 |**0.7109** | 0.7604 |**0.5433**|
> | 96 | 0.9216 |**0.5663**| 0.6704 |**0.4574**|
> | 48 | 1.0053 |**0.4832**| 0.7045 |**0.4172**|
>
> > **2. Why are no baseline results included? Without benchmarking, how should readers interpret the claimed improvements?**
>
> **Response:**
> Thank you for pointing this out. As mentioned above, standard FL baselines (e.g., FedAvg, FedProx) cannot be directly applied because they strictly require identical model architectures (and thus identical input dimensions) across clients.
>
> However, to provide a meaningful benchmark for interpretation, we have conducted additional experiments using the **"Linear Projection + FedPer"** strategy as a baseline. This represents the most straightforward alternative for dimension alignment. As shown in the comparison table in **Response 1**, these benchmarks confirm that FedQAPer provides substantial performance gains over the naïve alignment approach.
>
> > **3. How is feature heterogeneity simulated? Are the random feature splits representative of real-world multi-sensor or multimodal setups?**
>
> **Response:**
> We simulated feature heterogeneity by assigning each client a random subset of features, ranging from 1 to a maximum of $N$ features. We believe this setup is highly representative of real-world scenarios, particularly in **smart grids and energy management**.
>
> For example, in a building energy management system, different **distribution panels (or sub-meters)** monitor varying numbers of sub-loads depending on the specific zone or equipment they serve (e.g., HVAC, lighting, plug loads). Our random allocation strategy mimics this practical environment where the number and type of sensors vary significantly across different installation sites(clients).

---

> ### Author Response · Authors · 2025-11-21
> **Rebuttals (2/2)**
>
> > **4. Can you provide any empirical evidence of alignment (e.g., embedding similarity or visualization) to confirm that QAP actually harmonizes client feature spaces?**
>
> **Response:**
> We agree that empirical evidence of alignment is crucial to substantiate our claims. We will include visualizations of the latent representations processed by the QAP module.
>
> > **5. What is the communication and computational overhead introduced by QAP compared to FedPer?**
>
> **Response:**
> The QAP module is essential for bridging the dimensional gap among clients. As previously noted, a direct comparison to standard FedPer is structurally challenging because FedPer assumes identical input dimensions and cannot function in our heterogeneous setting.
>
> However, to quantify the overhead, we analyzed the ratio of QAP parameters to the total **shared parameters** transmitted during communication. As shown in the table below, the QAP module accounts for approximately **22-23%** of the total shared model size. We believe this moderate increase in communication cost is well justified, as it enables the federation of clients with diverse feature configurations.
>
> > **6. Does the framework generalize to partial feature overlap or clients with entirely disjoint features?**
>
> **Response:**
> Yes, the framework generalizes well to both partial overlap and entirely disjoint features. The QAP mechanism is designed to map whatever features a client possesses into the target latent space, regardless of whether those features overlap with others.
>
> To validate this, we applied FedQAPer to a real-world dataset collected from **non-residential building distribution panels**, where sub-loads are often disjoint. The model achieved robust performance in this setting, confirming its generalizability.
>
> > **7. Could QAP be integrated into other personalized FL schemes or is it somehow specific to FedPer-style architectures?**
>
> **Response:**
> Theoretically, QAP can be integrated with other FL schemes such as FedAvg to train the encoder/backbone, as it successfully unifies the input dimensions.
>
> However, we specifically chose a **FedPer architecture** because of the reconstruction requirement. Since each client needs to predict its original, unique number of features, a personalized output head is structurally necessary to map the shared backbone’s output back to the client-specific dimension. While QAP enables shared encoding, the personalized heads are critical for the prediction phase in this heterogeneous setting.
>
> > **8. Have you considered any real-world non-synthetic datasets where feature heterogeneity naturally occurs (e.g., multi-site sensor networks or IoT data)?**
>
> **Response:**
> Yes, we have extended our evaluation beyond synthetic splits. We applied FedQAPer to a **real-world dataset from non-residential building distribution panels**, where feature heterogeneity occurs naturally due to different circuit configurations.
>
> FedQAPer demonstrated superior performance compared to the simple projection baseline in this setting, verifying its practical applicability in real-world sensor network scenarios.
>
> > **9. Why does the validation loss across QAP dimensions using the iTransformer backbone jumps so suddenly for $d_{qap}=256$?**
>
> **Response:**
> We attribute this sudden increase in validation loss to **overfitting**. When $d_{qap}$ is increased to 256, it not only expands the latent representation size but also forces the backbone model’s internal hidden dimensions and the personalized heads to scale up accordingly.
>
> This results in a sharp increase in the total number of trainable parameters. Given the dataset size, this added complexity likely caused the model to overfit, leading to degraded generalization performance on the validation set compared to the optimal setting of $d_{qap}=128$.

---

### Comment · Area_Chair_xhvx · 2025-11-20
**To AI Review**

Dear authors,

I would like to share an important reminder regarding the review process. Recently, we have noticed that some reviewers may be using AI tools to help generate their reviews. This can lead to low-quality or inaccurate feedback, which is unfair to authors who deserve careful and thoughtful evaluations.

To help maintain fairness, I kindly ask for your assistance: If you believe a review you received was partly or fully generated by AI, and you have some evidence (for example: unusual writing style, clear factual mistakes, AI-detector results, repeated generic sentences, etc.), please feel free to contact me directly.

I will review any evidence you provide and, if appropriate, adjust the weight of the reviewer’s evaluation so that it does not negatively affect your submission. Thank you for helping us keep the review process fair and responsible. Your understanding and cooperation are greatly appreciated.

Best regards,

AC

---

### Comment · Area_Chair_xhvx · 2025-11-27

Dear Reviewers and Authors,

As we are approaching the rebuttal deadline, I would like to share a gentle reminder with everyone.

For authors:
If you have not yet submitted your rebuttal, please make sure to do so as soon as possible. Submitting very close to the deadline may reduce the chance for reviewers to read and respond in time, which could affect the discussion phase.

For reviewers:
If a rebuttal has already been submitted for your assigned paper, I encourage you to take a moment to read it and, where appropriate, provide a brief response or update your evaluation. Of course, this is not meant to pressure anyone into changing scores, it is simply to ensure that all reviews remain well-informed before final decisions.

Thank you all for your time and effort in keeping the review process smooth and constructive.

Warm regards,
AC

---

### Author Response · Authors · 2025-12-04
**Author Response to Reviews and Summary of Revisions**

Dear reviewers,

Thank you once again for reading our paper and providing valuable feedback on our work. We sincerely appreciate your time and effort in evaluating our research.
Before the final decision, we would like to summarize the key revisions made to our paper.

>**Motivation and Background**

Before describing the revisions, we would like to briefly clarify the motivation behind this research. This study originated from the problem of **predicting sub-loads in electrical distribution panels.** When each distribution panel is treated as a client, the number of sub-loads varies across panels, making it infeasible to feed them directly into a common backbone model. To address this, we introduced Query Attention Pooling (QAP). Furthermore, our goal was to improve prediction performance by enabling each panel to leverage not only its own information but also knowledge from other panels.

While existing federated learning research on zero-shot or few-shot prediction is certainly valuable, we believe that our research, which addresses the challenge of clients having varying numbers of features, is equally important and worth investigating.

---

>**Summary of Revisions**

**1. New training model (iSpikeformer)**
We introduced and trained a new SNN model called iSpikeformer, which is the SNN version of iTransformer, a model specialized for time-series forecasting. This addition enables an apple-to-apple comparison between ANN and SNN models.

**2. Inference energy comparison**
To demonstrate the efficiency of SNN models, we compared the energy consumption per inference between the ANN model (iTransformer) and the SNN model (iSpikeformer). iSpikeformer consumes approximately **113% less energy** than iTransformer.

**3. Simple projection baseline experiment**
To validate the effectiveness of the QAP module, we added experiments using a simple linear projection that only aligns the varying feature dimensions across clients to a unified dimension ($d_{qap}$). The results show that **QAP consistently outperforms simple projection across all time steps** when using the iSpikeformer model.

**4. QAP process visualization**
We added visualizations in the Appendix illustrating how raw data is transformed after passing through the QAP module.

**5. Code update**
We uploaded the updated code, including modifications for the simple projection baseline and iSpikeformer experiments.

**6. Minor revisions**
Based on feedback that the algorithm table was too complex, we simplified it for improved readability. We also adjusted font sizes and figure dimensions for better clarity.



Best regards,

The Authors

---

### Note · Program_Chairs · 2026-01-17
**Submission Desk Rejected by Program Chairs**

The following references in this submission do not refer to real documents and/or have major errors in bibliographic information:

 Wujie Wen, Xudong Ma, Jinjun Xiong, Wen-mei Hwu, and Yihang Tang. Differentiable spike: A backpropagation-friendly spiking neural network. In Advances in Neural Information Processing Systems, volume 36, 2023.